# The Development of Maillard Reaction, and Advanced Glycation End Product (AGE)-Receptor for AGE (RAGE) Signaling Inhibitors as Novel Therapeutic Strategies for Patients with AGE-Related Diseases

**DOI:** 10.3390/molecules25235591

**Published:** 2020-11-27

**Authors:** Chieh-Yu Shen, Cheng-Hsun Lu, Cheng-Han Wu, Ko-Jen Li, Yu-Min Kuo, Song-Chou Hsieh, Chia-Li Yu

**Affiliations:** 1Institute of Clinical Medicine, National Taiwan University College of Medicine, Taipei 10002, Taiwan; tsichhl@gmail.com (C.-Y.S.); b89401085@ntu.edu.tw (C.-H.L.); chenghanwu@ntu.edu.tw (C.-H.W.); 543goole@gmail.com (Y.-M.K.); 2Department of Internal Medicine, National Taiwan University Hospital, National Taiwan University College of Medicine, Taipei 10002, Taiwan; dtmed170@yahoo.com.tw; 3Department of Internal Medicine, Kaohsiung Medical University College of Medicine, Kaohsiung 80756, Taiwan

**Keywords:** advanced glycation end products, receptor for AGEs, Maillard reaction, N^ε^-(carboxymethyl)-lysine, N^ε^(carboxyethyl)-lysine, glyoxalase, AGE-related diseases, Maillard reaction inhibitor, AGE–RAGE axis signaling, metabolic memory

## Abstract

Advanced glycation end products (AGEs) are generated by nonenzymatic modifications of macromolecules (proteins, lipids, and nucleic acids) by saccharides (glucose, fructose, and pentose) via Maillard reaction. The formed AGE molecules can be catabolized and cleared by glyoxalase I and II in renal proximal tubular cells. AGE-related diseases include physiological aging, neurodegenerative/neuroinflammatory diseases, diabetes mellitus (DM) and its complications, autoimmune/rheumatic inflammatory diseases, bone-degenerative diseases, and chronic renal diseases. AGEs, by binding to receptors for AGE (RAGEs), alter innate and adaptive immune responses to induce inflammation and immunosuppression via the generation of proinflammatory cytokines, reactive oxygen species (ROS), and reactive nitrogen intermediates (RNI). These pathological molecules cause vascular endothelial/smooth muscular/connective tissue-cell and renal mesangial/endothelial/podocytic-cell damage in AGE-related diseases. In the present review, we first focus on the cellular and molecular bases of AGE–RAGE axis signaling pathways in AGE-related diseases. Then, we discuss in detail the modes of action of newly discovered novel biomolecules and phytochemical compounds, such as Maillard reaction and AGE–RAGE signaling inhibitors. These molecules are expected to become the new therapeutic strategies for patients with AGE-related diseases in addition to the traditional hypoglycemic and anti-hypertensive agents. We particularly emphasize the importance of “metabolic memory”, the “French paradox”, and the pharmacokinetics and therapeutic dosing of the effective natural compounds associated with pharmacogenetics in the treatment of AGE-related diseases. Lastly, we propose prospective investigations for solving the enigmas in AGE-mediated pathological effects.

## 1. Introduction 

Advanced-glycation end products (AGEs) are heterogeneous molecules derived from post-translational nonenzymatic modifications of macromolecules including proteins, lipids, and nucleic acids by glucose or other saccharides (fructose and pentose). AGEs are deleterious molecules and are found to be increased in the plasma of physiological aging and age-related diseases [1,2,3,4], diabetes mellitus [4,5,6,7], and autoimmune/inflammatory rheumatic diseases, including systemic lupus erythematosus [8,9], rheumatoid arthritis [10,11], systemic sclerosis [12,13], adult-onset Still’s disease [14], and psoriasis [15,16]. AGEs, by binding with their receptors of AGEs (RAGEs), can promote oxidative stress leading to the production of proinflammatory cytokines and inflammatory reactions [17]. In this context, oxidative stress can disturb intracellular signals to become pathological states, particularly insulin-mediated metabolic responses and insulin resistance [5,18].

AGE formation is induced by Maillard reaction (MR), named after French scientist Louis Camille Maillard for his pioneering work in 1912. He observed the yellow–brown color change when reducing sugar was heated with amino acids [19]. To date, more than 20 different AGEs have been identified in human blood and tissues, and in dietary foods. Arbitrarily, these AGEs can be classified into fluorescent and nonfluorescent [20,21]. The most important nonfluorescent components are carboxymethyl-lysine (CML), carboxyethyl-lysine (CEL), and pyrraline [22,23]; the most important fluorescent AGEs include pentosidine and methylglyoxal-lysine dimer (MOLD) [24,25]. The key characteristic feature of these AGEs relies on the presence of lysine residue in the molecules. Metabolically, the AGEs are catabolized in renal proximal tubular cells and are excreted from kidneys [26]. AGE accumulation can cause metabolic burden such as hyperglycemia and hyperlipidemia, oxidative stress, inflammatory responses, and endothelial dysfunction after binding with RAGEs [27].

Various environmental factors, including high-carbohydrate and high-calorie diets, high-temperature-cooked food, cigarette smoke, and a sedentary lifestyle may enhance AGE formation [28]. Many investigators suggested that AGEs are produced on “common soil” in metabolic diseases [29]. Accordingly, AGEs could become a striking link between modern diet and health [30]. We first briefly discuss the formation of AGEs by Maillard reaction, and then the factors affecting AGE formation, the deleterious effects by AGEs, the molecular bases of AGE-related diseases, the biomolecules and phytochemical compounds effective for AGE-related diseases, the metabolic memory, the “French paradox”, and the pharmacokinetics and therapeutic doses of effective natural compounds in association with pharmacogenomics are discussed in detail in the successive sections. 

## 2. Formation of AGEs by Maillard Reaction (MR)

In general, AGE formation is a relatively slow process in the physiological conditions of the body. Therefore, AGE deposition usually occurs in slow turnover rate protein molecules such as tissue collagens or lens crystalline. Briefly, the glycation of proteins is induced by MR in three steps: (1) the slow formation of Schiff bases, (2) the early formed unstable AGE precursors that may underlie Amadori rearrangement, and (3) the formation of late irreversible AGE products [31,32]. Schiff bases as early unstable AGEs are derived from condensation between the electrophilic carbonyl group of a reducing sugar with the free amino groups, especially lysine or arginine residues [29]. The next step is the rearrangement of unstable bases to the formation of stable ketoamines called Amadori products. Both compounds remain unstable and may react irreversibly with other peptides/proteins to form protein cross-links. These cross-linked proteins then undergo oxidation, dehydration, or polymerization to become a variety of stable AGEs [33,34]. Chemobiologically, AGEs can combine with metal ions such as Cu^2+^ or Fe^2+^ to provide catalytic sites for the generation of reactive oxygen species (ROS) and reactive nitrogen intermediates (RNI) [35,36,37].

The binding of AGEs with their multiligand immunoglobulin superfamily receptors, RAGEs, can induce oxidative stress, inflammatory response, and endothelial dysfunction [27]. However, other groups of cell-surface receptors of AGEs with opposite functions to RAGE, including AGE-R1, R2, and R3, and macrophage scavenger receptors are instead involved in AGE homeostasis [22,23]. These homeostatic receptors act as regulators of endocytosis and clearance after binding with noxious AGE molecules [38,39]. Vlassara et al. [26] demonstrated that inverse correlation existed between AGE-R1-to-RAGE ratio and the levels of oxidative stress in both humans and mice. Figure 1 depicts the formation and fates of AGEs after binding with different AGE receptors in the body. 

## 3. Factors Involved in Accelerating AGE Formation and Accumulation in the Body

Some endogenous and environmental factors may accelerate AGE formation in the body. These factors are discussed in the following subsections.

### 3.1. Endogenous Factors

#### 3.1.1. Hyperglycemia

A primary consequence of hyperglycemia is the formation of AGEs and induction of oxidative damage (OX). Perkins et al. [40] evaluated the effects of experimental hyperglycemia on OX and AGE–RAGE signaling in healthy obese individuals. They concluded that these obese but healthy persons could prevent the accumulation of AGEs and OX during metabolic stress by the mechanism of increasing the fractional excretion of AGEs via renal clearance. In contrast, Aragno et al. [41] demonstrated in animal experiments that high fructose consumption might lead to AGE accumulation in different tissue types in association with peripheral insulin resistance and deranged lipid metabolism. These data suggest that fructose-induced AGEs are deleterious to metabolic syndrome. However, healthy obese people rather than morbidly obese (MO) ones are more tolerant during metabolic stress.

#### 3.1.2. Aging, Oxidative Stress (OS), and Aging-Related Inflammatory Disease

The cause–effect relationship between AGEs and the aging process remains debatable. Some authors argued that the aging process accelerates the accumulation of AGEs [2,3,42,43]. Other authors suggested that AGE formation played a crucial role in the natural course of the aging process [40,41]. Nevertheless, circulating glycotoxins are undoubtedly linked to oxidative stress and inflammatory response leading to cell dysfunction [3,17,44]. Li et al. [42] found that the levels of cardiac AGEs were ~2.5-fold higher in aged hearts than those in young ones. In addition, a group of proteins with molecular weight of 50~75 kDa and isoelectric point of 4~7 were distinctively modified in aged hearts due to enhanced cardiac AGE accumulation, and AGE-enhanced OS and protein modifications by AGEs. Son et al. [44] demonstrated that the age-related accumulation of AGE-albumin, S100N, and the membrane expression of RAGE were more prominent in visceral rather than in subcutaneous fat. They concluded that visceral fat was involved in the pathogenesis of inflammation-induced complications in the elderly. Reynaert et al. [45], and Li et al. [46] overviewed age-related chronic inflammatory diseases including cardiovascular disease, Type 2 diabetes mellitus (Type 2 DM), chronic obstructive pulmonary disease, neurodegenerative diseases, and osteoporosis. The authors concluded that AGE–RAGE signaling disturbances were the common contributing factors for the inflammatory state in these noncommunicable chronic inflammatory diseases. 

#### 3.1.3. Obesity 

Obesity is usually associated with an increased risk of metabolic syndrome including insulin-resistance Type 2 DM, hypertension, fatty liver, and vascular complications attributed to the overproduction of adipokines from fat cells. Gaens et al. [47] reported that increased plasma and tissue levels of MGO, AGEs, and advanced lipoxidation end products (ALE) surrogated by CML were found in obesity. Brix et al. [48] demonstrated that patients with MO had significantly lower soluble-form RAGE (sRAGE) compared with the levels in the non-obese group. However, sRAGE levels increased after weight reduction by bariatric surgery that suppressed the AGE-mediated inflammation process. Similarly, Sanchez et al. [49] measured AGE accumulation by using skin autofluorescence (SAF) in the forearm with an AGE reader. The group noted that MO patients with metabolic syndrome had higher SAF levels than those of non-obese individuals. After bariatric surgery, SAF continued to remain high until glycemic memory failed. Deo et al. [50] evaluated the effect of weight loss on the CML levels of overweight participants without diabetes. CML values were decreased by 17% after weight loss, but this was less effective in patients with diabetes or prediabetes without being overweight. These results may suggest that both overweight and hyperglycemia contribute to AGE production and tissue accumulation in the body. The effect of hyperglycemia seems to be greater than that of overweight.

#### 3.1.4. Chronic Renal Insufficiency

Plasma levels of AGEs were found to be dramatically elevated in uremia patients with or without diabetes [51,52]. Miyata et al. [53] used an intravenous injection of a synthesized AGE, pentosidine, into rats to investigate the fate of AGEs. The group clearly demonstrated that pentosidine was filtrated by renal glomeruli, reabsorbed in the proximal renal tubules in the sites for catabolism or modifications, and excreted in urine. Later, Asano et al. [54] used three cell lines (proximal tubular, distal tubular, and nonrenal cell lines) to study the metabolism of protein-linked pentosidine. They demonstrated that pentosidine was rapidly detected in the cytoplasm of the proximal renal tubular cell line (TJC-12), but not the distal tubular (MDCK) or nonrenal (BALB3T3) cell line. They concluded that renal proximal tubular cells played an essential role in the disposal of plasma pentosidine. Following the same line, Waanders et al. [55] directly detected renal pentosidine accumulation in nondiabetic chronic adriamycin-induced nephropathy in rats. AGE accumulation in chronic kidney disease concomitantly led to the development of chronic heart failure, cardiovascular diseases, diabetes, neurodegenerative disease, osteoarthritis, and nondiabetic atherosclerosis [56]. The consumption of a high-AGE-containing diet may also be a risk factor for chronic diseases, including chronic renal disease [57]. This so-called “glycation stress” was found to be closely associated with kidney aging, as suggested by Inagi [58].

#### 3.1.5. Glyoxalase I Deficiency 

Miyata et al. [59] investigated the role of the glyoxalase detoxication system on reactive carbonyl compounds (RCOs), the precursors of pentosidine, in the AGE contents of a uremia patient on hemodialysis. The authors incidentally found that the plasma levels of pentosidine and CML were extremely high in the renal blood vessels (RBVs) of this patient rather than those of hemodialysis patients. Further investigation showed very low activity of glyoxalase in the RBVs of this patient. They concluded that glyoxalase I deficiency (GLO-I) failed to detoxify AGEs and played a crucial role in elevated AGE levels in uremia patients. In addition, Shinohara et al. [60] reported that the overexpression of GLO-I in bovine endothelial cells suppressed intracellular AGE formation and prevented the hyperglycemia-induced enhancement of macromolecular endocytosis in the blood. Similarly, Brouwers et al. [61] demonstrated that the overexpression of GLO-I reduced hyperglycemia-induced AGE production and oxidative stress in mesangial cells derived from diabetic rats and mice [62]. Furthermore, Kurz et al. [63] showed that GLO-I induction could decrease the toxic levels of MGO, GO, and other AGEs to protect cell damage from glycation stress. Xue et al. [64] explored the molecular basis of the transcriptional control of GLO-I by transcription factor erythroid 2-related factor 2 (Nrf2). The group thereby delineated a defense mechanism against decarbonyl glycation (methylglyoxal)-induced stress by GLO system in high glucose concentration, inflammation, cell aging, and senescence. Recently, Garrido et al. [65] reported that fatty acid synthesis could collaborate with GLO-I in protecting glycation toxicity to limit the cellular accumulation of MGO-derived AGEs. 

#### 3.1.6. Autoimmune and Inflammatory Reactions Facilitate AGE Formation 

It is well documented that the plasma levels of AGEs are elevated in autoimmune, rheumatic inflammatory [8,9,10,11,12,13,14,15,16,66], neuroinflammatory [67], neurodegenerative [63,68], and neuropsychiatric diseases [69], and cancers [70]. 

The inflammatory reaction activates innate immune-related cells including macrophages, dendritic cells, CNS-resident microglia cells and astrocytes, and induces a metabolic switch toward glycolysis in these inflammatory cells [71,72,73]. During glycolytic metabolism, MGO is synthesized from glyceraldehyde-3-phosphate and dihydroxyacetone phosphate, whereas GO is directly converted from glucose [74,75]. As a consequence, the production of MGO and GO becomes the precursor for AGE formation. The interaction of MGO with lysine residue leads to N^ε^-(carboxymethyl) lysine (CML) [76] formation. The interaction of GO with lysine leads to the N^ε^-(carboxyethyl) lysine (CEL) formation [76]. To protect the body from the toxic effects of increased MGO and GO, the methylglyoxalase and glyoxalase systems, which consist of GLO-1 and GLO-2, are activated to catabolize these two AGE precursors [77].

The molecular bases of AGE formation in neurodegenerative diseases such as Alzheimer’s (AD) and Parkinson’s (PD) are different from other inflammatory reactions in the brain tissue. Instead, α-synuclein deposition is thought to be crucial for AGE formation in situ in the two diseases. α-Synuclein is found to be associated with synaptic vesicles [78], and it acts as the chaperone of the SNARE complex, which is responsible for vesicle fusion and neurotransmitter release [79]. Moreover, the deposition of α-synuclein suppresses GLO-I expression and results in the impairment of AGE clearance [63]. 

### 3.2. Environmental Factors

Undoubtedly, diet-derived AGEs are major contributors to the body’s AGE pool and cardiometabolic disorders [80]. However, there are many modifiable factors in diet AGEs, such as contents, cooking temperature, and personal lifestyle variations (sedentary habit or exercise); cigarette smoking profoundly affects AGE production and tissue accumulation [18]. 

#### 3.2.1. Effect of Different Diets on AGE Formation

Davis et al. [81], and Lopez-Moreno et al. [82] conducted a pilot study to determine whether a high-fat diet could influence plasma AGE levels. They concluded that there was no effect of dietary fat on AGE formation. Later, Chhabra et al. [83] confirmed that repeated cooking-oil heating could contribute to increased AGEs in diet. Simple lime addition could reduce AGEs in fried foods. Furthermore, Kim et al. [84] confirmed that a diet high in red and processed meat with refined grains significantly increased plasma CEL compared to an energy-matched diet that was high in whole grains, dairy, nuts, and legumes. 

The Mediterranean diet (Med diet) is rich in monounsaturated fatty acids and minimally processed natural foods, and is found to reduce postprandial oxidative stress and inflammation [85]. Following this line, Lopez-Moreno et al. [86] demonstrated that the Med diet could reduce serum AGE levels and enhanced antioxidant defenses in the elderly. The same authors further found that the Med diet supplemented with the Q_10_ coenzyme enhanced redox parameters, decreased AGE levels, and increased AGE-R1 and GLO-1 mRNA [87]. 

#### 3.2.2. Cigarette Smoking Accelerates Toxic AGE Formation

In addition to unavoidable UV light, ionizing radiation, and air pollution in the environment [28], cigarette smokers have long been found to have an increase in cardiometabolic complications. Accordingly, the possibility of tobacco components in initiating the formation of glycation, especially after burning, has been suspected. Cerami et al. [88] confirmed that glycotoxins were transferred from cigarette combustion to the serum proteins of human smokers. Therefore, plasma AGE-apolipoprotein B and AGE-albumin levels in cigarette smokers were significantly higher than those in nonsmokers. Nicholl et al. [89] further found an increase in AGEs in the lung and blood vessels of cigarette smokers. For further understanding the underlying chemicobiological mechanism, Dickerson et al. [90] identified that nornicotine, a constituent of tobacco and metabolite of nicotine, could catalyze aldol reactions under aqueous conditions and caused aberrant protein glycation. The same group also showed higher plasma nornicotine-modified protein glycation in smokers. Prasad et al. [91] found that cigarette smoke induced AGE, but reduced sRAGE formation, consequently resulting in the development of cardiovascular complications. Federico et al. [92] indirectly confirmed that skin AGE accumulation in breastfed infants from smoking mothers was higher than that from nonsmoking mothers.

Factors affecting aberrant AGEs formation are shown in Table 1. 

## 4. Cellular and Molecular Bases of AGE–RAGE axis Induce Signaling in Inflammation, Microvascular Endothelial Cell, Smooth Muscle Cell, and Fibroblast Damage, and Renal Podocyte Functional Impairment in AGE-Related Diseases

The binding of AGEs with different cell-surface-expressed AGE receptors activates different signaling pathways in cells. The most studied AGE receptor, RAGE, is a multiligand transmembrane receptor that belongs to the immunoglobulin superfamily with crucial roles in chronic inflammatory response and immune dysfunction [31]. In addition to binding with AGEs, RAGEs can also bind with diverse ligands including high-mobility group box-1 (HMGB-1), s100/calgranulins, Mac-1 and amyloid-β peptide [93]. Recent investigations revealed several other AGE receptors that were successively identified, consisting of the AGE receptor complex (AGE-R1/OST-48, AGE-R2/80k-H, AGE-R3/galectin-3) and scavenger receptor family (SR-A, SR-B, SR-1, SR-E, LOX-1, FEEL-1, FEEL-2 and CD36) [31], as previously shown in Figure 1.

AGE–RAGE interaction transduces signals via four pathways: (1) JAK-2-STAT 1, (2) PI_3_K–AKT, (3) MAPK–ERK, and (4) NADPH oxidase–ROS. Lastly, the phosphorylated NF-κB enters into the nucleus to transcribe the gene expression of proinflammatory cytokines, growth factors, profibrotic cytokines, and oxidative stress [31,93,94,95]. Accordingly, AGE–RAGE interaction transduces inflammatory and fibrotic signals in immune-related cells to induce somatic cell damage, and lastly tissue fibrosis in various tissue types and organs, as shown in Figure 2. 

The pathological effects of AGEs on different immune-related cells with released molecules with regard to tissue damage are discussed individually in the following subsections. 

### 4.1. Effects of AGE–RAGE Interaction on Innate and Adaptive Immune Responses

#### 4.1.1. Effects of AGE on Polymorphonuclear Neutrophil (PMN) Functions 

The most important deleterious effect of AGEs in DM patients is the development of oxidative stress, which is associated with the functional derangements of many cells and tissue types. Gupta et al. [96] demonstrated that AGE–PMN interaction upregulated NADPH oxidase expression, and enhanced ROS and RNI generation, resulting in microvascular endothelial cell damage. Bansal et al. [97] directly used AGE–HSA to investigate the effects on PMN functions. Their results showed that AGE–HSA enhanced the ROS and RNI production of PMN by the upregulation of NADPH oxidase and inducible nitric oxide synthase (iNOS). Recently, Lu et al. [98] found that AGEs activated the neutrophil release of myeloperoxidase (MPO) and elastase (NE), and deranged CD_4_+T cell differentiation via these two granular proteins. Alterations in CD_4_^+^T cell differentiation by AGEs were found as suppression in both Th1 (denoted by IFN-γ) and Th17 (denoted by IL-17) phenotypes, and changes in the expression of transcription factor T-bet in Th1, RORγt in Th17, and FoxP_3_ in Treg cells.

#### 4.1.2. AGE Effects on Inflammatory Responses of Macrophages and Immune Dysfunctions of Helper T Lymphocytes

Van der Lugt et al. [99], and Byum et al. [100] demonstrated that dietary AGEs could induce TNF-α secretion from human macrophage-like cells and activated macrophages to generate more AGEs. This suggests that AGEs could act as an accelerator to induce inflammatory response via macrophage activation. Shen et al. [101] showed that AGE–BSA significantly enhanced IL-6 production from monocytes/macrophages, but suppressed Th1 (IL-2 and IFN-γ) and Th2 (IL-10) gene expression. Enhanced monocytic IL-6 production was proven via MAPK-ERK and MyD88- transduced NF-κB p50 signaling pathways. Our results are somewhat different from those of Lu et al. [98], in that they argued Th1 gene expression was enhanced by AGEs. However, they did not determine the effects of AGEs on Th2 cytokine production. The real cause of this discrepancy between the two studies remains to be elucidated. 

#### 4.1.3. AGE Effects on Human Fibroblasts

Diabetes is a risk factor for the occurrence of periodontal diseases and the progression of periodontitis. Nonaka et al. [102] found that AGEs increased IL-6 and ICAM-1 expression in human gingival fibroblasts via interactions with RAGE and transduction of the MAPK-NF-κB signaling pathway. These inflammatory reactions exacerbated the progression of periodontal disorders. Dai et al. [103] isolated fibroblasts from human diabetic wounds, and found that these AGE-exposed fibroblasts increased cell apoptosis after activating NLRP3 inflammasomes via an ROS-induced signal pathway. In summary, AGE can enhance both the inflammatory reaction and cell apoptosis of fibroblasts via the ROS-inducing capacity to impair fibroblast functions. 

#### 4.1.4. AGE Effects on Aging-Related Macular Degeneration (AMD)

AMD is one of the major vision-threatening diseases of the elderly and DM patients. Anand Babu et al. [104] proved that AGEs acted as a pro-oxidant metabolite to induce pro-inflammatory cytokine IL-6, IL-8, and vascular endothelial cell growth factor (VEGF) release from human retinal pigment epithelial cells. These cytokines could facilitate macrophage infiltration through oxidative stress, inflammation, chemotaxis, and neovascularization to cause macular degeneration. 

The pathological effects of AGE–RAGE interaction in immune-related cells, fibroblasts, and retinal pigmented cells in mediating AGE-related pathology are summarized in Figure 3. 

On the basis of these observations, the pathological effects of AGEs on diabetes-associated major complications, including cardiovascular diseases and nephropathy, are discussed in more detail in the following subsections.

### 4.2. AGE Effects on Microvascular Endothelial and Smooth Muscle Cell Damage in Diabetic Cardiovascular Disease (CVD)

The main cause of mortality in diabetic patients is CVD. Recently, increased attention has been focused on the mechanism of endothelial cell damage in CVD development in DM patients. An experiment was conducted to compare the effects of high- and low-AGE meals on macro- and microvascular endothelial functions, and oxidative stress in Type 2 DM patients by Negrean et al. [105]. Data revealed that a high-AGE meal induced a more acute deleterious effect on vascular functions than a low-AGE meal did. Yamagishi et al. [106] demonstrated that the cross-talk between the AGE–RAGE axis and the dipeptidyl peptidase-4 (DPP4)–incretin system were intimately involved in the development and progression of diabetes-associated complications. These complications include diabetic microangiopathy, arteriosclerotic CVD, Alzheimer’s disease, and osteoporosis. The DPP4–incretin system contains glucagon-like peptide-1 (GLP-1) and glucose-dependent insulinotropic polypeptide (GIP-1). Both gut hormones are secreted from the intestine in response to food intake to augment insulin release and suppress glucagon secretion. Therefore, the DPP-4–incretin system can rapidly suppress GLP-1 and GIP-1 secretion, and subsequently enhances the development of diabetic vascular complications. 

One of the long-term complications of kidney transplantation is CVD. Liu et al. [107] attempted to identify the noxious molecules and the molecular mechanism of kidney transplantation-induced arteriosclerosis. The authors found that increased serum AGE levels enhanced the production of α-smooth muscle actin (α-SMA) and osteopontin (OPN) expression in the vascular smooth muscles (VSMC), contributing to arteriosclerotic changes in these patients. In rat experiments, augmented expression of α–SMA, OPN, and integrin-linked kinase (ILK) in vascular smooth muscle cells was prominent after AGE treatment. The authors concluded that AGEs played a crucial role in arteriosclerosis after renal transplantation by activating VSMC-to-osteoblast trans-differentiation via the AGE–RAGE–ILK pathway. These observations were further confirmed by the findings of Xu et al. [108] that the mTOR inhibitor, sirolimus, reduced AGE-mediated arteriosclerosis in kidney-transplantation recipients via suppression on the ILK/mTOR pathway. Furthermore, AGE could induce the proliferation and migration of vascular smooth muscle cells through the PI_3_K/AKT pathway [109]. VSMC could also be induced to become foam cells by CML that further accelerated vascular calcification in diabetic patients [110]. 

Another paradoxical phenomenon of diabetic vascular disease is its long-term persistence, even after hyperglycemia has been controlled well for a while. This may be attributed to the so-called “metabolic memory” under the mechanism of ongoing oxidative stress [111] and epigenetic changes [112]. El-Osta et al. [111] found that transient hyperglycemia could induce the long-lasting activation of epigenetic changes in the promoter site of the NF-κB p65. This may subsequently enhance monocyte chemoattract protein 1 and vascular cell adhesion molecule 1 expression to sustain vascular inflammation. These proinflammatory cytokine gene expressions could be prevented by reducing mitochondrial superoxide radical production. Furthermore, Aschner et al. [112] found that metabolic memory might become permanent epigenetic-change association with the activation of histone modifications to provoke inflammatory gene expression. This is because AGE–RAGE signaling can induce proinflammatory cytokine expression and oxidative stress via the NF-κB pathway, as shown in Figure 2, similar to hyperglycemia-induced inflammation and epigenetic changes. In conclusion, both hyperglycemia and AGE molecules can mediate long-term “metabolic memory” found in diabetic patients with CVD complications. 

It remains worthwhile to discuss the effects of the renin–angiotensin–aldosterone system (RAA) in diabetic CVD complications. Scheen et al. [113] discovered that RAA system inhibition could prevent Type 2 DM. Kintscher [114] proved that irbesartan, an angiotensin-receptor blocker, could simultaneously treat patients with both hypertension and metabolic syndrome. Recently, Cabandugama et al. [115] demonstrated that the RAA system was involved in cardiorenal and metabolic syndrome.

The metabolic and molecular mechanisms of AGEs in inducing diabetic cardiovascular complications are illustrated in Figure 4.

### 4.3. AGE Effects on Diabetic Nephropathy (DN)

Neuropilin-1(NRP-1) is a transmembrane glycoprotein defined as a receptor for members of the semaphorin family. NRP-1 was found in the neuronal vascular system and in kidney podocytes. Reduced NRP-1 expression is characteristic for DN [116]. Bondeva et al. [117,118,119] showed that AGEs suppressed NRP-1 expression in kidney podocytes in association with decreased podocyte migration and adhesion by reducing the binding ability of specificity protein 1, Sp1, a transcription factor to NRP-1 promoter. Yuan et al. [120] induced a diabetic rat model by intraperitoneal injection of high-fat/-sucrose diet and a low dose of streptozocin. They found that CML elicited DN via disturbing the intracellular feedback regulation of the cholesterol metabolism, thereby increasing renal lipid accumulation. 

The molecular and cellular pathogenesis of AGEs-induced diabetic nephropathy is shown in Figure 5.

## 5. Novel Biomolecules and Phytochemical Compounds Acting as Maillard Reaction Inhibitors, Preformed AGE Breakers, and AGE–RAGE Signaling-Pathway Blockers

AGEs contribute to the development of physiological aging and many major chronic diseases, including diabetic pathology, and neurodegenerative, autoimmune/inflammatory, and metabolic cardiovascular diseases. Accordingly, it is valuable to search for novel AGE inhibitors, including AGE formation inhibitors, preformed AGE breakers, and AGE–RAGE signaling blockers as potential therapeutic interventions for AGE-related diseases. The underlying modes of action of different AGE inhibitors are based on the attenuation of glycosylation, antioxidative stress, metal ion chelating, and scavengers of reactive 1,2-dicarbonyl compounds or ROS/RNI [121]. Arbitrarily, these novel therapeutic AGE inhibitors can be classified into 4 categories: (1) inhibitors of AGE formation; (2) breakers of preformed AGEs; (3) blockades of AGE–RAGE axis signaling; and (4) inducers of intracellular glyoxalase, ubiquitin–proteasome, and autophagy pathways [121,122,123,124,125]. The next subsections discuss in detail novel AGE inhibitors in the four abovementioned categories published only in recent years (after 2015). 

### 5.1. Plasma Amines, Including Amino Acids and Peptides Acting as Protein-Glycation Inhibitors

Chilukuri et al. [126] extensively reviewed the literature and found that some plasma amino acids could inhibit glycation by hampering the binding of glucose to protein molecules by competitive inhibition. These amino acids included positively charged amino acids (l- and d-lysine, arginine) by contributing their ability to scavenge glyoxal and methylglyoxal. On the other hand, negatively charged amino acid aspartic acid contributes by inhibiting AGE formation. Sulfur-containing amino acid taurine was shown to inhibit Schiff base formation between proteins and reactive sugars. Cysteine was capable of inhibiting protein glycation, trapping dicarbonyl compounds, and activating the glycoxalase system. 

Naturally occurring dipeptide l-carnosine (β-alanyl-l-histidine) can inhibit AGE formation by quenching reactive carbonyl species, and it is capable of affecting transglycation. A dipeptide isolated from fresh garlic, γ-glutamyl-S-allyl-cysteine (GSAC), can attenuate Maillard reaction during the initial and late stages of glucose-induced protein denaturation. The antiglycation activity of the two other dipeptides derived from fresh garlic, γ-glutamyl-methylcysteine (γ-GMC) and γ–glutamyl-propylcysteine (γ-GPC), was attributed to their scavenging property. 

### 5.2. Dipeptidyl Peptidase 4–Incretin Inhibitors for Interrupting AGE–RAGE Signaling Pathway

As discussed in Section 4.2, a pathological cross-talk exists in diabetic pathology between the AGE–RAGE and DPP4–incretin axes [106]. In animal experiments, DPP4 deficiency or the addition of a DPP4 inhibitor could protect against experimental diabetic nephropathy in a glucose-lowering- independent manner [127].

### 5.3. DNA Aptamers Acting as Preformed AGE Breakers

Aptamers are a group of short, single-stranded DNA or RNA molecules that can bind with high affinity/specificity to a wide range of target proteins [95,127,128]. Yamagishi et al. [129] demonstrated that specific DNA aptamers raised against AGEs could bind and break AGE activity, and consequently blocked AGE–RAGE axis signaling. These specific DNA aptamers can become novel therapeutic molecules for various AGE-related diseases. 

### 5.4. Exogenous Soluble-Form RAGE (sRAGE) in Amelioration of AGE–RAGE Signaling Pathway

sRAGEs are produced either from alternative gene splicing or proteolytic cleavage of membranous RAGEs. In a rat model of Type 2 diabetic nephropathy, sRAGE administration was confirmed to diminish albuminuria and glomerulosclerosis [130]. In addition, exogenous administration of sRAGEs could be helpful in the amelioration of diabetes-related CVD [131]. 

Table 2 summarizes the effective biomolecules and phytochemical compounds acting as Maillard reaction inhibitors, preformed AGE breakers, and AGE–RAGE signaling blockers with their modes of action in the treatment of AGE-related diseases. 

In recent years, an increasing number of biomolecules and phytochemicals have been successively discovered. We now focus on the protection of DM-related complications, individually divided into nephropathy, cardiovascular diseases, and retinopathy in the following subsections.

### 5.5. Novel Biomolecules and Phytochemical Compounds Potentially Protecting from Diabetic Nephropathy 

DN is one of the most important microvascular complications in Type 1 and 2 diabetic patients where undoubtedly, AGEs play a pathological role. AGE–RAGE axis activation transduces various intracellular signaling pathways, including PI_3_K/AKT, MAPK/ERK, JAK2–STAT1, and NADPH oxidase to provoke oxidative stress and chronic inflammation in various renal-cell components (Figure 2). Thereby, blocking AGE–RAGE axis signaling would be beneficial for diabetic nephropathy [95]. In the following subsections, the therapeutic strategy and working mechanism of different phytochemicals in protecting different renal-cell components are discussed.

#### 5.5.1. Therapeutic Intervention on AGE-Induced Podocyte Functional Impairment 

Renal glomerular podocytes are specialized epithelial cells that play a crucial role in maintaining the glomerular filtration barrier. A unique structure of podocytes relies on long arborized interdigitating protrusion fingers to create a filtration system called “slit diaphragm” or “slit junction”. These long protrusions of podocytes require coordination between actin skeleton and cell membrane that involve several key proteins in forming a protein complex. Ezrin is a pivotal cross-linker molecule within this complex. McRobert et al. [132] found that AGEs suppressed ezrin activity in renal proximal tubular cells (Figure 5). Overexpression of ezrin in differentiated human immortalized podocytes completely reversed AGE–BSA, impairing the migration and adhesion capacity of podocytes. 

Eucalyptol is a natural component of aromatic plants with antioxidant activity. Kim et al. [133] found that the oral administration of eucalyptol in db/db diabetic mice enhanced the expression of slit-diaphragm proteins α-actinin-4 and integrin β_1_ in diabetic kidneys with the enhancement of the foot process and the amelioration of glomerular fibrosis. The molecular basis of this improvement is mediated by the blockade of ERK-C-Myc-enhanced nephrin and CD2AP expression in AGE-exposed podocytes. 

Plasma D-ribose levels are elevated in Type II DM. It seems possible that increased blood D-ribose is involved in diabetic nephropathy since D-ribose can induce AGE formation. Hong et al. [134] discovered that long-term administration of D-ribose in C57BL/6J mice stimulated NLRP3 inflammasome formation and was activated in murine kidney podocytes via the AGE–RAGE signaling pathway, leading to glomerular injury. It is advised to decrease uptake of fructose or ribose to prevent inflammation in patients with Type 2 DM.

#### 5.5.2. Therapeutic Intervention on AGE-Induced Mesangial, Glomerular Endothelial, and Myofibroblastic Cell Dysfunction 

Berberine (C_20_H_18_NO_4_^+^) is an isoquinoline alkaloid isolated from *Coptidis rhizome* and *Cortex phellodendri*. This photochemical compound can improve fasting blood sugar, body weight, renal functions, and histopathological changes in diabetic kidneys. Qiu et al. [135] demonstrated that berberine inhibited mesangial-cell proliferation in vitro via decreased intracellular AGE, RAGE, p-PKC, and TGF-β1 levels. The authors concluded that the protective effect of berberine in diabetic nephropathy was through the inhibition of the AGE–RAGE–PKCβ–TGFβ1signaling pathway. 

Glomerular endothelial dysfunction may lead to the progressive deterioration of the renal architecture and functions in early DN stages. Salvianolic acid A (SalA) is a water-soluble phenolic acid extracted from the dry root and rhizome of *Salvia multiorriza Bunge*. Investigations demonstrated that SalA exhibited anti-ischemic, anti-inflammatory, and anti-diabetic effects that are especially beneficial for anti-endothelial dysfunction. Hou et al. [136] treated SalA in diabetic nephropathy rats and found that SalA restored glomerular endothelial permeability through the rearrangement of actin cytoskeleton by way of the AGE–RAGE–RhoA/ROCK pathway. Moreover, SalA attenuated AGE-induced oxidative stress, and the subsequent inflammation and distended autophagy of glomerular endothelial cells via suppressing the AGE–RAGE–Nox4 axis. 

In the long course of chronic inflammation, AGEs play a causative role in kidney fibrosis via the induction of matrix protein deposition in tissue. Chrysin is an active flavone present in passion flowers, honey, and mushrooms, with anticancer, antioxidant, anti-inflammatory, and neuroprotective effects. Kang et al. [137] reported that chrysin suppressed diabetes-associated renal tubulointerstitial fibrosis via blocking epithelial-to-mesenchymal transition (EMT). This unique phytochemical could also ameliorate podocyte damage by inhibiting endoplasmic reticulum stress. The same group then demonstrated that chrysin attenuated the accumulation of myofibroblast-like cells and matrix proteins in renal glomeruli of db/db diabetic mice [138]. 

#### 5.5.3. Therapeutic Intervention on Nrf2/ARE/GLO-1 Pathway

As mentioned above, MGO and GO are the most potent reactive carbonyl species and precursors for AGE formation. The glyoxalase (GLO) system contains two enzymes, GLO-I and GLO-II, for the detoxification of MGO and GO; thereby, it inhibits AGE formation. Nuclear factor erythroid 2-related factor 2 (Nrf2) is an essential component of the antioxidant responsive element (ARE) to transcribe antioxidant enzymes and GLO-I gene expression. One of the major causes for the downregulation of GLO-I expression in diabetes is the downexpression of Nrf2. Do et al. [139] explored the pharmacological effects of *Eucommia ulmoides* (EU) extract, a medicinal herb commonly used in Asia to treat hypertension and diabetes. They found that EU significantly upregulated Nrf2 but downregulated RAGE expression. Immunohistological analysis of diabetic kidneys revealed a reduction in AGEs and MGO accumulation in tissue after EU treatment for 6 months. In addition, a significant increase in the protein amount and enzymatic activity of GLO-I was noted in the blood of patients with DN. The same group also evaluated another medicinal herb, *Spatholobus suberectus* (SS), commonly used in patients with anemia, menoxenia, and rheumatism [140]. Results revealed that SS significantly upregulated GLO-1 and NADPH quinine oxidoreductase 1 (NQO1) expression, but conversely reduced CML accumulation and RAGE expression. These results suggest that SS ameliorated renal damage by inhibiting diabetes-induced glycotoxicity and oxidative stress via the Nrf2/ARE/GLO-1 pathway.

Recently, Chen et al. [141] found that hesperetin, a plant flavonoid largely derived from citrus (sweet orange and lemon), ameliorated pathological process in diabetic rats via the Nrf2/ARE/GLO-1 pathway.

The biomolecules and phytochemicals beneficial for diabetic nephropathy are listed in Table 3 with their working mechanisms. 

### 5.6. Therapeutic Intervention of Novel Biomolecules and Phytochemical Compounds on Diabetes-Related CVD

AGE–RAGE-mediated CVD can be ameliorated by certain novel therapeutic interventions. Traditionally, AGE formation in DM patients can be inhibited by hypoglycemic medications, vitamins, and quitting the smoking of cigarettes. Prasad et al. [131] also reviewed the literature, and concluded that statins, telmisartan, and curcumin could inhibit RAGE expression, whereas statins, ACE inhibitors, rosiglitazone, and vitamin D enhanced RAGE levels. 

Dhar et al. [142] conducted a cell-based experiment and found that high-glucose and high-MGO-treated rat cardiomyocytes expressed high amounts of caspase-3, BAX, RAGE, NF-κB, and ROS. The use of AGE cross-link breaker alagebrium (ALA) could attenuate MGO and AGE formation in rat H9C2 cardiac myocytes. 

Matsui et al. [143] demonstrated that sulforaphane derived from myrosinase-treated glucoraphanin, a widely found phytochemical in cruciferous vegetables (broccoli, kale, or cabbage) could effectively inhibit inflammation in AGE-exposed HUVECs and AGE-infused rat aorta through the suppression of RAGE expression. 

Furthermore, naturally occurring dipeptide L-carnosine (also see Section 5.1), which abundantly exists in skeletal muscle and other excitable tissue types, attenuated fasting blood sugar, triglycerides, AGEs, and TNF-α levels in patients with Type 2 diabetes [144]. Obviously, supplementation of L-carnosine is beneficial for diabetes-related CVD. 

Sanchis et al. [145] identified myoinositol hexaphosphate (phytate; IP6), a natural phytochemical compound abundant in cereals, legumes, and nuts, which exhibited the capacity to chelate cationic metals and thereby inhibited metal-catalyzed protein glycation. This evidence indicates that the dietary supplement with IP6 can potentially prevent the development of diabetes-related CVD. 

Other phytochemicals against AGE–RAGE axis signaling were critically reviewed by Yamagishi et al. [146], who revealed that quercetin, sulforaphane, iridoids, and curcumin were protective for endothelial cells, blood vessels, and against heart damage in diabetes-related CVD. However, caution should be taken of the “metabolic-memory” effect in treating chronic abnormalities in diabetic CVD, as mentioned in Section 4.2. [112]. Yamagishi et al. [147] further emphasized the importance of the “metabolic memory” effect in sustaining chronic abnormalities in diabetes vessels. It is not easily reversed, even if hyperglycemia has been controlled for 10–30 years. This long-term memory can explain why former cumulative diabetic exposure could contribute to the current progression of diabetic vascular complications. 

### 5.7. Therapeutic Intervention on Diabetic Retinopathy (DR)

DR is a serious microvascular complication characterized by increased retinal angiogenesis, enhanced capillary permeability, pericyte loss, capillary basement membrane thickening, and retinal detachment in the retinal layer [148]. Elevated levels of AGEs were found to be closely correlated with DR severity [149]. Lee et al. [150] identified a novel phytochemical, 5′-methoxybiphenyl -3,4,3′-triol (K24), isolated from *Osteomeles schwerinae* C.K. Schneid, which exerted an inhibitory effect on AGE rat serum albumin-induced retinal vascular leakage after intravitreal injection into rat eyes. K24 also significantly reduced retinal nonperfused areas and neovascular tufts in oxygen-induced retinopathy. The molecular basis of the antiglycation and antiangiogenic activities of K24 relies on its suppressive effects on vascular endothelial growth factor (VEGF) production and decreasing the loss of occludin. 

Shao et al. [151] demonstrated that (-)-epicatechin, a monomeric flavonoid found in high concentrations of cocoa and green tea, was able to effectively trap MGO. Furthermore, Kim et al. [152] discovered that the exogenous injection of (-)-epicatechin into rats could improve AGE accumulation in retinas via its AGE-breaking capacity. 

Parveen et al. [153] extensively reviewed medicinal-plant-derived phytochemicals and found that these novel natural phytochemicals could target multiple pathological factors, including ROS, AGEs, AGE–RAGE signaling pathways, and hexosamine flux. These activities could improve AGE-induced retinopathy. 

Table 4 lists the chemicals and novel phytochemicals that are effective in the treatment of diabetic cardiovascular complications and retinopathy with their modes of action.

Although some clinical trials have been conducted with AGE breakers and AGE–RAGE signaling inhibitors, including L-carnosine [145,154], RAGE-Aβ inhibitor [155], and DPP4 inhibitor combined with PPAR-γ agonist [156], no satisfactory results could be obtained and applied in clinical practice till now. 

The term “French paradox” began to appear in 1992 to describe the relatively low incidence of CVD in the French population, despite a relatively high dietary intake of saturated fats in the food. This paradox was attributable to the light to moderate consumption of red wine [157]. Resveratrol and other polyphenolic compounds derived from red wine were found capable of decreasing oxidative stress, increasing nitric oxide bioavailability, enhancing cholesterol efflux from vessel walls, inhibiting lipoprotein oxidation, and suppressing macrophage cholesterol accumulation and foam-cell formation [158]. Thereby, resveratrol can become a potential therapeutic in neuroprotection [159], cardiovascular diseases [160], rheumatic joint disorders [161], diabetic cardiomyopathy [162], and renal fibrosis [163]. However, a complex foodome contains >25,000 substances in the human diet [164] and more than 4000 varieties of flavonoids have been identified in fruits, vegetables and beverages such as tea and wine. The working mechanisms of flavonoids depend on anti-oxidative effects, direct radical scavenging, enhancing xanthine oxidase, inhibiting leukocyte immobilization, and interacting with other enzyme systems [165]. In addition to these beneficial effects, flavonoids also exhibit adverse effects that vary largely among individuals, dependent upon genetic variations in their pharmacokinetics and pharmacodynamics [166]. Accordingly, the doses of different natural products should be optimized to improve efficacy and reduce toxicity according to pharmacogenetics. These pharmacogenetic properties can determine drug metabolic enzymes, drug transporters, and interactions with the pharmacological targets or pathways of the individual responses to the natural products.

## 6. Conclusions and Perspectives

AGE–RAGE axis activation in immune-related and somatic cells transduces signals for oxidative stress and inflammatory reactions that facilitate AGE formation and are involved in AGE-related diseases. These inflammatory reactions damage vascular endothelial cells, renal mesangial/endothelial/smooth muscle/podocytic cells, and retinal pericytes to cause diabetic triopathy. In addition to hyperglycemia, some environmental factors, such as high dietary AGE content, Western-style foods, cigarette smoking, and the Mediterranean diet may affect AGE formation. Patients with renal proximal tubular damage retard the catabolism and clearance of preformed AGE due to glyoxalase deficiency. A number of biomolecules and phytochemicals isolated from vegetables, legumes, fruits, or flavonoids, acting as AGE formation inhibitors, preformed AGE breakers, AGE–RAGE axis blockers, or glyoxalase stimulators, were only confirmed effective in cell-base or animal studies. These compounds are expected to become novel therapeutic agents in addition to traditional anti-hyperglycemic and anti-hypertensive drugs. Nevertheless, these biomolecules and phytochemicals should prove to be more effective and less side effects than traditional hypoglycemic therapy before clinical applications.

The following prospective investigations are suggested for solving the enigmas in the pathological roles of AGE:(1)the molecular mechanism of “metabolic memory” in AGE-mediated CVD;(2)the molecular mechanism of inflammation-enhanced AGE formation;(3)the molecular basis of AGE-induced Th_1_/Th_2/_Th_17_/Treg subpopulation aberration in the immune dysfunction of patients with AGE-related diseases;(4)elucidation of the pathological and pathogenic roles of hemoglobin A_1_C in patients with diabetes mellitus;(5)the glycosylation status and pathological roles of advanced-glycation end products of lipid and nucleic acid in vitro and in vivo; and(6)the pathogenic roles of AGE molecules on telomere shortening and telomerase activity in the physiological aging processes.

## Figures and Tables

**Figure 1 molecules-25-05591-f001:**
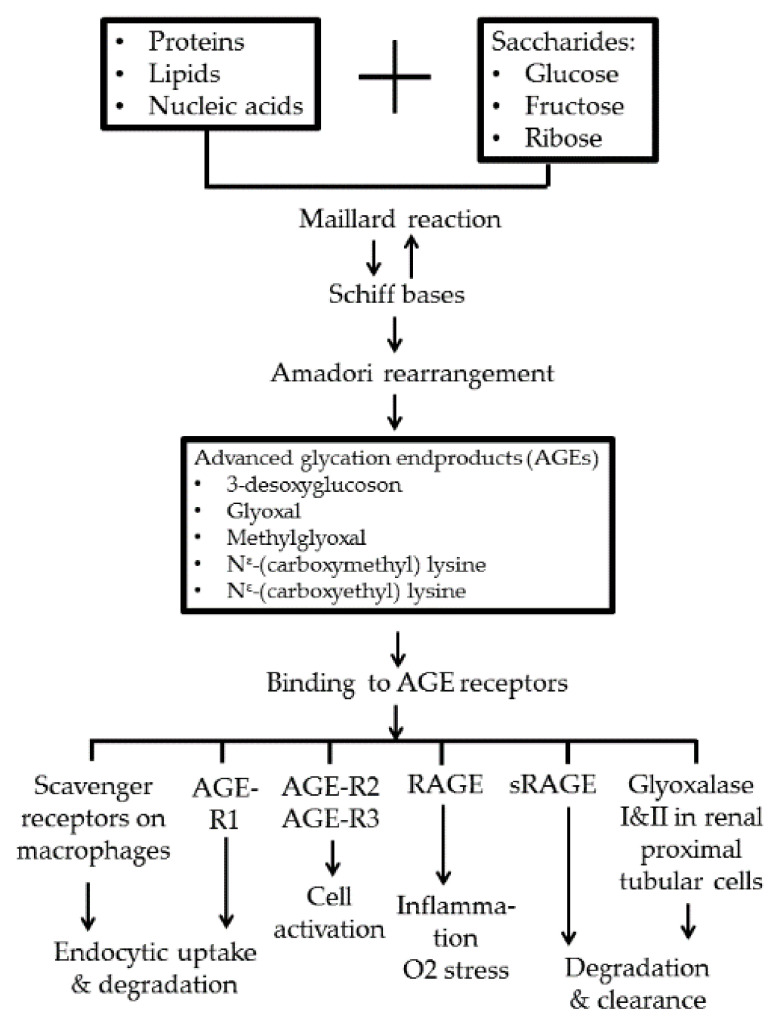
Advanced-glycation end-product (AGE) formation by Maillard reaction and the fates after binding with different AGE receptors. AGE-R1, R2, and R3: AGE receptors R1, R2, and R3; RAGE: receptor for AGE; sRAGE: soluble-form RAGE.

**Figure 2 molecules-25-05591-f002:**
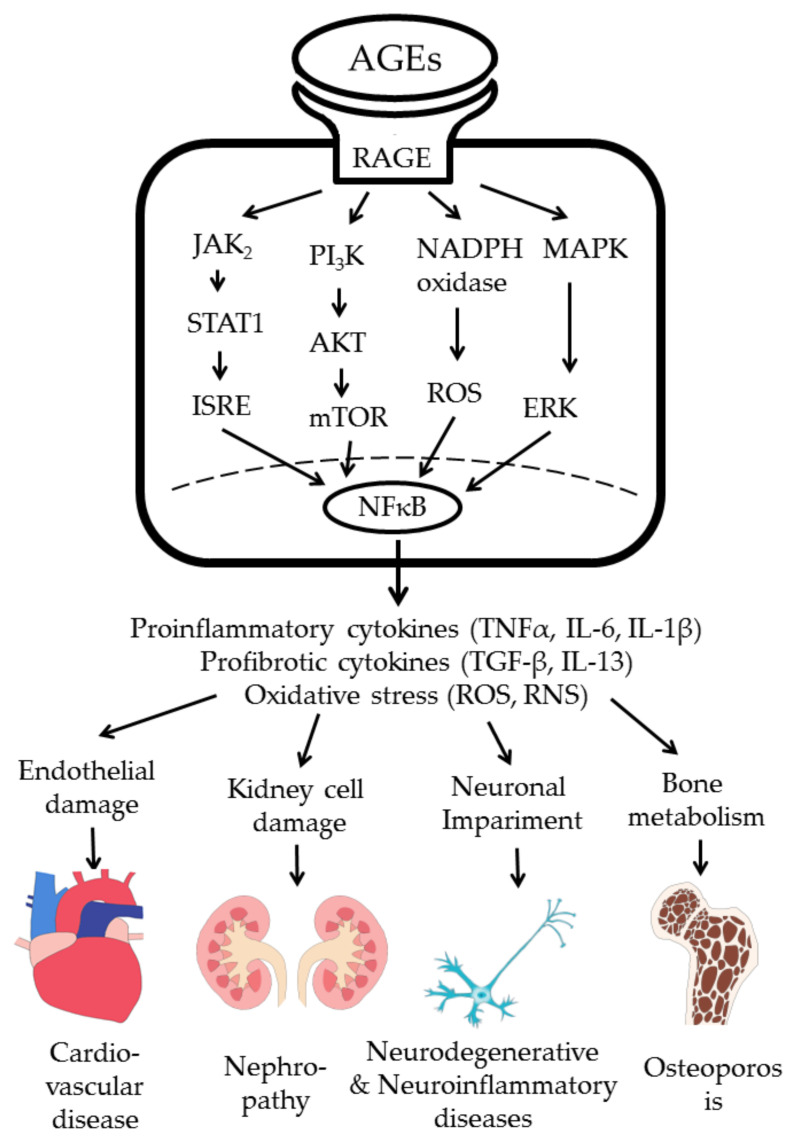
AGE–RAGE interaction in immune-related cells transduces signals for gene expressions of proinflammatory and profibrotic cytokines, and increased oxidative stress to induce cell and tissue damage. ISRE: interferon stimulated response element; ROS: reactive oxygen species; RNI: reactive nitrogen intermediates.

**Figure 3 molecules-25-05591-f003:**
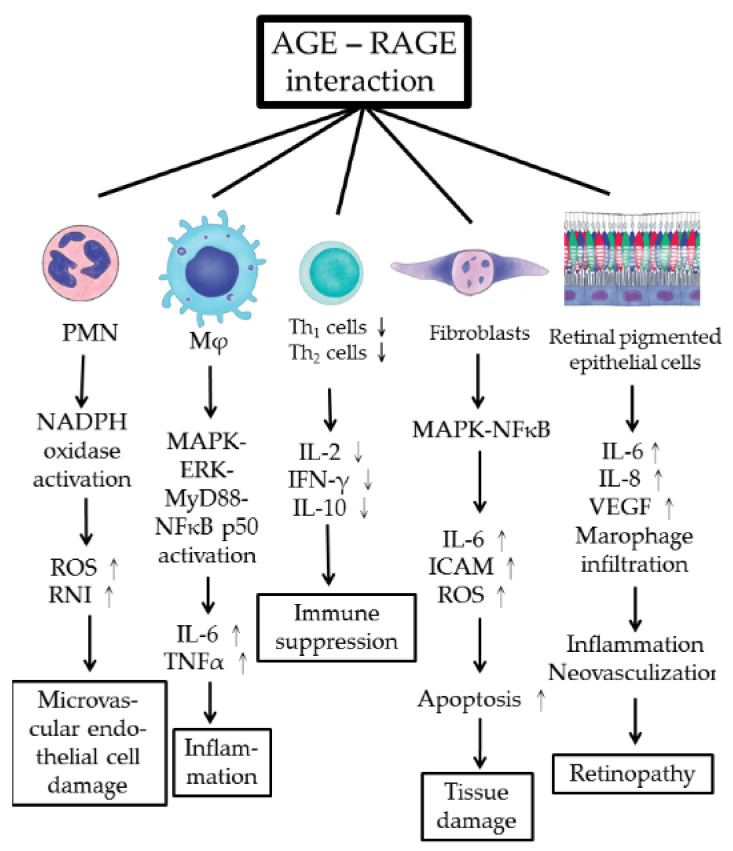
Cellular and molecular pathogenesis of AGE–RAGE axis activation in inducing microvascular endothelial cell damage, tissue inflammation, immune dysfunction, tissue fibrosis, and retinopathy. RPEC: retinal pigmented epithelial cell; ROS: reactive oxygen species; RNI: reactive nitrogen intermediates; ICAM: intercellular adhesion molecule; VEGF: vascular endothelial cell growth factor.

**Figure 4 molecules-25-05591-f004:**
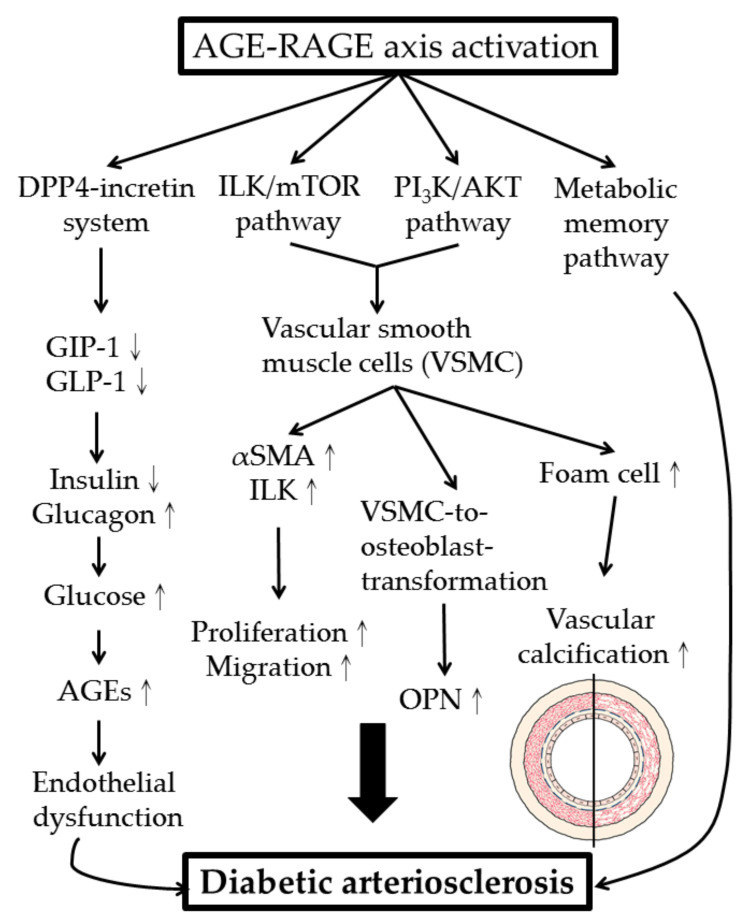
Metabolic and molecular mechanisms of AGE–RAGE axis activation in development of diabetic arteriosclerosis. DPP4: dipeptidyl peptidase 4; GIP-1: glucose-dependent insulinotropic polypeptide-1; GLP-1: glucagon-like peptide-1; VSMC: vascular smooth muscle cell; ILK: integrin-linked kinase; α-SMA: α-smooth muscle actin; OPN: osteopontin.

**Figure 5 molecules-25-05591-f005:**
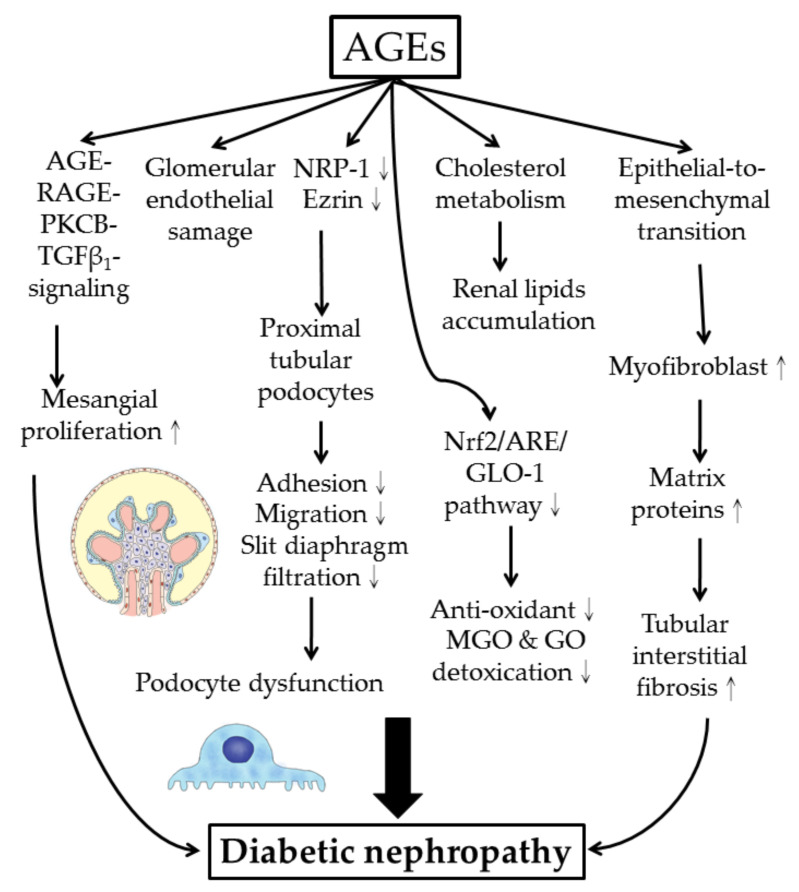
Cellular and molecular bases of AGE-mediated diabetic nephropathy. NRP-1: neuropilin-1; Nrf2: nuclear factor erythroid 2-related factor 2; ARE: anti-oxidant responsive element (for induction of anti-oxidant enzymes and glyoxalase 1 expression); MGO: methyglyoxal; GO: glyoxal.

**Table 1 molecules-25-05591-t001:** Factors affecting advanced glycation end product (AGE) formation.

○ **Endogenous factors:**
Aging [1,2,3,4]
Hyperglycemia [4,7,40,41]
Obesity [48,49]
Autoimmune and inflammatory reactions [8,9,10,11,12,13,14,15,16,17,66]
Chronic renal insufficiency [51,52,55,56,57]
Glyoxalase I and II deficiency [59,60]
Oxidative stress and chronic inflammation [71,72,73,74,75,76]
○ **Environmental factors:**
Dietary AGEs [28,30]
UV light and ionizing radiation [28]
Air population [28] Cigarette smoking [88,89,90,91,92]

**Table 2 molecules-25-05591-t002:** Biomolecules and phytochemicals acting as inhibitors of Maillard reaction and AGE–RAGE signaling in treatment of AGE-related diseases.

Molecule	Mode of Action
**[I] Plasma amino acids** [126]	Prevents glucose–protein binding
Positively charged: L- and D-lysine, arginine	─ Scavenger of glyoxal and methylglyoxal
Negatively charged: Aspartic acid	─ AGE formation inhibitor
Sulfur-containing: Taurine Cysteine	─ Schiff base formation inhibitor Trapper of dicarbonyl compounds and inducts glyoxalase system
**[II] Peptides** [126]	
Natural dipeptide: β-alanyl-L-histidine (L-carnosine)	─ Quenches reactive carbonyl species
Fresh garlic scale compound: γ-glutanyl-S-allyl-cysteine	─ Maillard reaction inhibition in initial and late stage of glucose-induced protein denaturation
Purified from fresh garlic γ-glutanyl-methylcysteine γ-glutanyl-propylcysteine	─ Scavenger activity Scavenger activity
**[III] Dipeptodyl-4-incretin axis inhibitors** [106,127]	─ Inhibits AGE–RAGE–DPP4–incretin cross-talk
**[IV] DNA aptamers** [95,127,128,129]	─ AGE formation and AGE–RAGE axis brokers
**[V] Soluble form RAGEs (sRAGEs)** [123,124]	─ AGE–RAGE signaling pathway inhibitor

**Table 3 molecules-25-05591-t003:** Novel phytochemicals and medicinal extracts with different anti-AGE activity in protecting from diabetic nephropathy.

Target Cell	Mode of Action
**[I] Renal tubular podocytes:**	
Phytochemical eucalyptol [133]	─Block ERK-C-myc enhanced nephrine and CD2AP expression induced by AGE.
	─Increase slit diaphragm protein α-actinin-4 integrin β_1_ and number of foot processes.
**[II] Glomerular mesangial cells:**
Phytochemical berberine [135]	─Decrease mesangial-cell proliferation by inhibiting intracellular AGE–RAGE–pPKC–TGFβ signaling pathway
**[III] Glomerular endothelial cells:**
Phytochemical salvianolic acid A [136]	─Increase rearrangement of actin cytoskeleton via AGE–RAGE–RhoA/ROCK pathway to restore glomerular endothelial permeability.─Reduce AGE-induced oxidative stress and inflammation.─Restore endothelial-cell autophagy via inhibition on AGE–RAGE–NOX4 axis.
**[IV] Tubulointerstitial fibrosis:**
Phytochemical chrysin [137]	─Suppresses myofibroblast-like cells and matrix protein accumulation via blocking epithelial-to-mesenchymal transition.─Reduces podocyte damage via decrease in endoplasmic reticulum stress.

**[V] Nrf2/ARE/GLO-1 pathway:**
Medicinal *Eucommia ulmoides* extracts [139]	─Upregulate Nrf2 and downregulate RAGE to enhance GLO-1 enzymatic activity.
Medicinal *Spatholobus suberectus* extracts [139,140]	─Upregulate GLO-1 and NADPH quinine oxidoreductase to reduce CML accumulation and RAGE expression.
Medicinal hesperetin [141]	─Enhance Nrf2/ARE/GLO-1 pathway.

**Table 4 molecules-25-05591-t004:** Novel medicines and phytochemicals acting as new therapeutic strategies in treatment of diabetes-induced cardiovascular disease and retinopathy.

Molecules or Phytochemicals	Mode of Action
**Diabetic cardiovascular disease:**	
[1] Statins, telmisartan, and phytochemical curcumin [131]	─ Inhibit RAGE expression
[2] Statins, AGE inhibitors, rosiglitazone, vitamin D [131]	─ Enhance sRAGE formation
[3] Alageberium (AGE crosslink breaker) [142]	─ Attenuates MGO and AGE formation in rat cardiomyocytes
[4] Phytochemical sulforophane [143]	─ Suppresses RAGE expression in AGE-exposed rat aorta and HUVEC
[5] Dipeptide L-carnosine [144]	─ Attenuates blood sugar, TG, AGE, and TNF-α levels in Type 2 DM
[6] Phytochemical myoinositol hexaphosphate [145]	─ Chelates cationic metal-catalyzed protein glycation
[7] Phytochemical quercetin, sulforaphane, iridoids, and curcumin [146]	─ Protect endothelial cells, blood vessels, and heart
**Diabetic retinopathy:**	
[1] Phytochemical 5′-methoxyliphenyl- 3,4,3′-triol [150]	─ Suppresses AGE-induced retinal vascular leakage ─ Suppresses VEGF production to prevent neovascularization ─ Decreases eye occluding loss.
[2] Phytochemical (-)-epicaterchin [152,153]	─ Traps MGO ─ AGE breaker

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
