# Peer review of "The Development of Maillard Reaction, and Advanced Glycation End Product (AGE)-Receptor for AGE (RAGE) Signaling Inhibitors as Novel Therapeutic Strategies for Patients with AGE-Related Diseases"

_molecules, 2020, doi:10.3390/molecules25235591_

Round 1

Reviewer 1 Report

Overall, this manuscript is very tiring and exhaustive to read. Within the first page, alone, there were extensive grammatical, syntax, and tense errors. Additionally, references were missing. Below is a brief list. I cannot justify continuing to edit the document. The authors will need to complete it.

Line 42: grammatical error- “found increase” should be “found to be increased”?

Line 46: grammatical error- reactions

Line 47: grammatical error- pathological states

Lines 51-54: Reference needed

Line 58: grammatical error- AGE accumulation

Line 68: grammatical error- AGE formation is

Lines 69-70: Incomplete sentence

Line 70: Define MR

Line 72: grammatical error: rearrangement, and (3)

Lines 82-84: Reference needed

*Abbreviations need to be defined.

Figures and Tables:

Table 1 is messy. Formatting, alignment, and bullets are confusing and of poor quality for a simple table

Figure 2 has (...) included in the text. Why not include the names of he cytokines and stressors? This is unacceptable in current state. Formatting and alignment are needed.

Consider re-evaluating and editing all tables and figures. The appearance of tables and figures is not of high quality.

Scientifically, the review is accurate and sound. The authors do a good job presenting the material in a logical progression. Of note, current primary literature should be included and cited in reviews and not other review articles. As this is a changing area, the authors should invest time in reviewing primary literature and not rehashing other reviews.

Author Response

Answers to the Reviewers and Editor

Thanks to Reviewers and Editor for offering us the opportunity to revise our manuscript (ID: molecules-973103). We took each comment and query seriously and tried our best to revise them point-to-point. The revised sites have been underlined for easy recognition in addition to the “tract change” function. We have also added 11 new references for more completion. The 4 tables have been reformatted and the 5 figures have also been redrawn for more vivid. The grammatical and typing errors have been corrected by MDPI English editing. We cordially hope the revised manuscript will be more acceptable by this world famous biochemical and molecular journal.

[I] Answers to the Reviewer (1):

 Comment (1): Overall, this manuscript is very tiring and exhaustive to read.

 Answer: We are extremely sorry for bothering you so much in reading the manuscript. We hope after the MDPI English editing, the revised manuscript will be easier to read by you and the readers.

 Comment (2): Within the first page, alone, there were extensive grammatical, syntax, and tense errors.. Below is a brief list. I cannot justify continuing to edit the document. The authors will need to complete it.

Line 42: grammatical error- “found increase” should be “found to be increased”?

Line 46: grammatical error- reactions

Line 47: grammatical error- pathological states

Line 58: grammatical error- AGE accumulation

Line 68: grammatical error- AGE formation is

Lines 69-70: Incomplete sentence

Line 72: grammatical error: rearrangement, and (3)

 Answer: We have already corrected them. The other typing and grammatical errors in text have also be corrected by ourselves and then by MDPI English editing.

 Comment (3): Additionally, references were missing including lines 51-54 and lines 82-84 in old edition.

 Answer: We have added two new references [20, 21] in P.2 line 52, and another 2 references [22, 23] in P.2 line 53.    .

Comment (4): Abbreviations need to be defined.

Answer: The abbreviation MR has been defined in the title of Section 2 and also in P. 19 of the “Abbreviation Section”

Comment (5): Table 1 is messy. Formatting, alignment, and bullets are confusing and of poor quality for a simple table.

Answer: We have to apology for this messy and distorted Table 1 due to format conversion from “Word” to the printed version. It is totally our mistake and carelessness. We have corrected it in the revised version.

Comment (6): Figure 2 has (...) included in the text. Why not include the names of the cytokines and stressors? This is unacceptable in current state. Formatting and alignment are needed.

Answer: Many thanks again for pointing out these big mistakes. We have deleted the (…) and added IL-1b for more accuracy.

Comment (7): Consider re-evaluating and editing all tables and figures. The appearance of tables and figures is not of high quality.

Answer: In obedience to your suggestion, we have already reformatted the 4 tables and redrawn the 5 figures for more accuracy and fineness.

Comment (8): Scientifically, the review is accurate and sound. The authors do a good job presenting the material in a logical progression. Of note, current primary literature should be included and cited in reviews and not other review articles. As this is a changing area, the authors should invest time in reviewing primary literature and not rehashing other reviews.

Answer: Thanks for this constructive comment. We have already cited the original article such as references 20, 22, 23, 111, 113-114 in the revised version to fulfill the requirement by the Reviewer.

Looking forward to hearing from you soon.

Yours sincerely,

Chieh-Yu Shen, MD

Chia-Li Yu, MD, PhD

Department of Internal Medicine

National Taiwan University Hospital

Taipei, Taiwan

Reviewer 2 Report

The review paper "The Development of Maillard Reaction and Advanced Glycation End Products (AGEs) -Receptor for AGE (RAGE) Signaling Inhibitors as Novel Therapeutic Strategies for Patients with AGE-related Diseases". It is well organized and I consider that it has had an exhaustive work on the subject.
1- I consider that mention should be made of the action of various antihypertensive mediations such as renin-angiotensin-aldosterone inhibitors and glitazone.
2- What clinical studies are currently being carried out AGE inhibitors or RAGE antagonists?
3- In relation to the microvascular complications of diabetes mellitus, what concept does metabolic memory deserve, and what relationship may exist with AGEs?

Author Response

Answers to the Reviewers and Editor

Thanks to Reviewers and Editor for offering us the opportunity to revise our manuscript (ID: molecules-973103). We took each comment and query seriously and tried our best to revise them point-to-point. The revised sites have been underlined for easy recognition in addition to the “tract change” function. We have also added 11 new references for more completion. The 4 tables have been reformatted and the 5 figures have also been redrawn for more vivid. The grammatical and typing errors have been corrected by MDPI English editing. We cordially hope the revised manuscript will be more acceptable by this world famous biochemical and molecular journal.

[II] Answers to the reviewer (2):

Comment (1): The review paper "The Development of Maillard Reaction and Advanced Glycation End Products (AGEs) -Receptor for AGE (RAGE) Signaling Inhibitors as Novel Therapeutic Strategies for Patients with AGE-related Diseases". It is well organized and I consider that it has had an exhaustive work on the subject.

Answer: Many thanks for the positive comment to inspire us from the Reviewer.

Comment (2): I consider that mention should be made of the action of various antihypertensive mediations such as renin-angiotensin-aldosterone inhibitors and glitazone.

Answer: Very appreciation for this constructive suggestion. We have added a paragraph in P. 10 lines 351-355 in response to this suggestion that:

         It remains worthy to discuss the effects of renin-angiotensin-aldosterone system (RAA) in diabetic CVD complications. Scheen et al. [113] discovered that RAA system inhibition could prevent thpe 2 DM. Kintscher [114] proved that irbesartan, an angiotensin receptor blocker, could simultaneously treat patients with both hypertension and metabolic syndrome. Recently, Cabandugama et al. [115] demonstrated that RAA system involved in cardiorenal and metabolic syndrome.”

 “

Comment (3): What clinical studies are currently being carried out AGE inhibitors

        or RAGE antagonists?

Answer: This is also an important suggestion to increase the academic and clinical values of this review article. We have only cited the recent clinical reliable trials after 2010 in P.18 lines 570-573 in response to this suggestion that;

          “Although some clinical trials have been conducted with AGE breakers and AGE-RAGE signaling inhibitors including L-carnosine [154, 155], RAGE-Ab inhibitor [156], and DPP4 inhibitor combined with PPAR-g agonist [157], no satisfactory results could be obtained and applied in clinical practice till now.”

Comment (4): In relation to the microvascular complications of diabetes mellitus,

   what concept does metabolic memory deserve, and what relationship may exist with AGEs?

  Answer: The concept of “metabolic memory” is utmost important in the treatment of diabetic-related cardiovascular complications. In addition to hyperglycemia, AGE molecule per se may also induce proinflammatory and pro-fibrotic cytokine gene expression those may also involve the epigenetic changes and subsequent metabolic memory. A paragraph have been added in P. 10 lines 337-350 that;

          El-Osta et al. [111] found that transient hyperglycemia could induced long-lasting activation of epigenetic changes in the promoter of the NF-κB p65. This may subsequently enhanced monocyte chemoattractant protein P and vascular cell adhesion molecule 1 expression to sustain the vascular inflammation. These proinflammatory cytokine gene expression could be prevented by reducing mitochondrial superoxide radical production. Furthermore, Aschner et al. [112] have found that the metabolic memory may become permanent epigenetic-changes association with the activation of histone modifications that provoke inflammatory gene expression. This is because AGE-RAGE signaling can induce pro-inflammatory cytokine expression and oxidative stress via NF-kB pathway as shown in Figure2, similar to the hyperglycemia-induced inflammation and epigenetic changes. In conclusion, both hyperglycemia and AGE molecules can mediate long-term “metabolic memory” found in diabetic patients with CVD complications.”

Looking forward to hearing from you soon.

Yours sincerely,

Chieh-Yu Shen, MD

Chia-Li Yu, MD, PhD

Department of Internal Medicine

National Taiwan University Hospital

Taipei, Taiwan

Round 2

Reviewer 1 Report

There has been very little improvement regarding edits. The abstract and introduction are still poorly written. 

When describing previous work, past tense needs to be used. 

The "French Paradox" should lead into your discussion of the pharmacokinetics and therapeutic doses. This is still an introduction to resveratrol.

The authors make blatantly inaccurate statements. Below are some examples.

Line #258: "The results of this study indicate that RV activates SIRT" does not make sense. Authors failed to link prior findings to statement regarding activation of SIRT proteins through TGFB/Smad3 pathway. More details should be included to make connection as TGFB is linked to fibrosis.

Line #265: Fibroblasts are not the largest population of cells in the heart.

Line #276: Diabetes mellitus is not one of the most important fibrotic diseases of the heart.

These are just a few. Please review extensively.
